# Impacts of the COVID-19 pandemic on the spatio-temporal characteristics of a bicycle-sharing system: A case study of Pun Pun, Bangkok, Thailand

Tawit Sangveraphunsiri[1☯], Tatsuya Fukushige[2☯], Natchapon Jongwiriyanurak[3☯], Garavig Tanaksaranond[4☯], Pisit Jarumaneeroj[1,5☯]*

1 Faculty of Engineering, Department of Industrial Engineering, Chulalongkorn University, Bangkok, Thailand, 2 Institute of Transportation Studies, University of California, Davis, California, United States of America, 3 SpaceTimeLab, Department of Civil, Environmental and Geomatic Engineering, University College London, London, United Kingdom, 4 Faculty of Engineering, Department of Survey Engineering, Chulalongkorn University, Bangkok, Thailand, 5 Regional Centre for Manufacturing Systems Engineering, Chulalongkorn University, Bangkok, Thailand

☯ These authors contributed equally to this work.
* pisit.ja@chula.ac.th

**Data Availability Statement:** All relevant datasets are provided on the following site: https://github.com/Maplub/odtensor.

## Abstract

The COVID-19 pandemic is found to be one of the external stimuli that greatly affects mobility of people, leading to a shift of transportation modes towards private individual ones. To properly explain the change in people's transport behavior, especially in pre- and post- pandemic periods, a tensor-based framework is herein proposed and applied to Pun Pun–the only public bicycle-sharing system in Bangkok, Thailand–where multidimensional trip data of Pun Pun are decomposed into four different modes related to their spatial and temporal dimensions by a non-negative Tucker decomposition approach. According to our computational results, the first pandemic wave has a sizable influence not only on Pun Pun but also on other modes of transportation. Nonetheless, Pun Pun is relatively more resilient, as it recovers more quickly than other public transportation modes. In terms of trip patterns, we find that, prior to the pandemic, trips made during weekdays are dominated by business trips with two peak periods (morning and evening peaks), while those made during weekends are more related to leisure activities as they involve stations nearby a public park. However, after the first pandemic wave ends, the patterns of weekday trips have been drastically changed, as the number of business trips sharply drops, while that of educational trips connecting metro/subway stations with a major educational institute in the region significantly rises. These findings may be regarded as a reflection of the ever-changing transport behavior of people seeking a sustainable mode of private transport, with a more positive outlook on the use of bicycle-sharing system in Bangkok, Thailand.

**Funding:** TS is supported by the Second Century Fund (C2F), Chulalongkorn University, Bangkok, Thailand (https://www.research.chula.ac.th/the-second-century-fund-chulalongkorn-university-c2f). The funder had no role in study design, data collection and analysis, decision to publish, or preparation of the manuscript.

**Competing interests:** The authors have declared that no competing interests exist.

# 1. Introduction

Since the emergence of the Coronavirus Disease 2019 (COVID-19) in December 2019, COVID-19 has become a major health threat that affects the whole world. This is especially evident from the number of confirmed COVID-19 cases that continuously increased and recently topped 500 million cases on April 14, 2022 –about 15 months after it reached the 100 million milestone on January 25, 2021 [1].

In order to keep the pandemic under control, a wide range of non-pharmaceutical mitigation strategies, such as lockdown and work-from-home measures, have been accordingly imposed, coupled with worldwide vaccination programs [2]. Although such measures have proven themselves useful in fighting against the COVID-19 outbreak, as they help reduce the number of daily contacts–and so risks of infection–they adversely affect mobility of people and their respective transport behavior [3–6]. Particularly, people are more concerned in taking crowded public transportation [7–11], leading to a shift of transportation modes towards private individual ones [12–16]. While safer, this new transportation pattern, however, deteriorates not only the efficacy of public transportation systems but also their long-term sustainability–which might be worse off after the end of COVID-19 outbreak, where the pre-pandemic level of transportation activities resumes [17].

To better address this issue, many have explored the development of more private but yet sustainable modes of transportation that meet the ever-changing mobility behavior of people (see [10,18–20], for example). And, bicycle sharing is among the alternative modes of green transportation that has been highlighted worldwide. In terms of pandemic-related responses, bicycling could help travelers limit their exposure with others, while avoiding crowded places in urban areas due to its flexibility [21,22]. In this regard, Jobe and Grif [23] found that the pandemic positively affected the use of bicycle-sharing system in San Antonio, Texas, and the respondents tended to continue using such a system in a longer term. Their findings were in-line with the works by [22,24], where the adoption rate of bicycle sharing in Slovakia cities and Beijing rose with longer trip durations during the COVID-19 pandemic. Bicycle sharing was also found to be more resilient in terms of service recovery when compared with other modes of transportation, as illustrated by [21,25–27].

Apart from such benefits, a city can further gain several advantages from bicycle sharing. For instance, it supports a healthier means of transportation, but with less transportation costs [28–31]. It also makes public transportation more accessible [32–34] and less congested [35–37], especially in downtown areas. Notwithstanding such a fact, bicycle sharing is largely adopted as an alternative means of transportation by urban residents of European countries, the United States, and China, mainly because of their sufficiently good bicycling infrastructure and legislation [38–40]. Besides these countries, bicycling is merely a means of leisure activities [41–47] for people with relatively-high socio-economic status [48], despite attempts made by many local authorities to develop more efficient bicycle-sharing systems in complement with their current transportation networks [49]. These failures are due largely to lack of understanding in patron's behavior and the roles of bicycle-sharing systems in the underlying transportation networks that may also vary from one to another setting, depending on a number of endogenous and exogenous factors, including trip patterns and patron's behavioral changes caused by external stimuli (*e.g.* the COVID-19 pandemic) [50].

To better address this issue, while supporting the growth of this sustainable mode of transportation in a longer term, a tensor decomposition-based framework has been herein adapted and applied to Pun Pun–the only public bicycle-sharing system in Bangkok, Thailand–in both pre- and post-pandemic periods so that the effects of COVID-19 pandemic on the spatio-temporal characteristics of Pun Pun could be quantified. To this end, the information of bicycle-

sharing trips from 2018 to 2020 is first collected; and, it is then decomposed into elements according to Pun Pun's relevant characteristics–namely, time-of-day, day-of-week, origin, and destination modes–by a non-negative Tucker decomposition approach. With this analytical framework, all related players would be able to comprehend the current state of Pun Pun and its role in the transportation network of Bangkok, which is of paramount importance to the development of such a system alongside others in a more sustainable fashion. It also helps support regulators in the development of adaptive actions, such as network alterations, as patron's behavior is proven to be vulnerable to external stimuli.

The remaining of this paper is organized as follows. Section 2 provides a discussion of transportation in Bangkok, Thailand, together with its corresponding COVID-19 situation, while Sections 3 and 4 entail data collection process and the proposed methodology. All results and analyses are then provided in Section 5. Finally, Section 6 concludes all the work and possible research directions.

## 2. Study area and background information

### 2.1. Transportation in Bangkok

Bangkok, the capital city of Thailand, is a place of work for nearly 16 million people, including the residents of five nearby provinces who commute to Bangkok on a daily basis. According to the survey by the Office of Transport and Traffic Policy and Planning (OTP) [51], in 2017, more than 33 million trips were made daily within the Bangkok Metropolitan Region (BMR); and, it was estimated that such a number would rise to 40 million trips by the end of 2042. Yet, nearly half (43%) of these trips were made by private cars, followed by private motorcycles (26%), while public transportation accounted only for 20% of all the trips. Among these trips, the number of bicycle trips was also found limited, despite supports from Bangkok residents, due largely to weather conditions and lack of proper bicycling infrastructure [52–54].

The tropical climate of Bangkok is among the most fundamental factors that negatively impacts the use of bicycling [46,55], as Bangkok is hot and humid throughout the year. The average temperature in Bangkok is around 30˚C (86˚F), but it could rise to 40˚C (104˚F) during the summer months, *i.e.* mid-February to May [56]. Although the weather during winter months, *i.e.* from mid-October to mid-February, is milder and suitable for bicycling, air pollution problems typically arise, with significantly high level of fine particulate matter (PM 2.5) that prevents commuters from bicycling on a regular basis.

In terms of infrastructure, bicycling has received much less attention in Thailand, when compared to other modes of transportation. It was even scarcely contained in the Land Transport Acts (B.E. 2522/ A.D. 1979) due to its perceived function as a recreational activity, coupled with Bangkok's heavy reliance on automobiles. This tendency was, however, changed after the cabinet approval for bicycling supports on November 19, 2013. In 2014, the National Non-Motorized Transport Plan (NMT plan) was proposed for the first time [53], followed by several nationwide bicycling campaigns, such as Bike for Mom and Bike for Dad in 2015. These campaigns prominently helped bicycling gain more public attention as a physical boost activity and an alternative means of transportation in a daily life at the same time [57,58]. To better promote bicycling as a safe, sustainable, first- and last-mile mode of transportation, the Thai government also included bicycling into the 12[th] National Economic and Social Development Plan (12[th] NESDP 2017–2021).

Stipulated by these developments, the OTP and Bangkok Metropolitan Authority (BMA) have thence designed a bicycle network in Bangkok–mostly based on those of bicycling events in the past couple years–initially for tourism purposes [59]. Later on, the infrastructure of these routes will be gradually expanded in accordance with Bangkok Metropolitan's 20-year

Development Plan (2013–2032)–with an overall goal to achieve a share of 8% by 2032 [60]. Despite the BMA's plans to bring forth better bicycling experience, the development of bicycling infrastructure in Bangkok, unfortunately, falls significantly behind expectations, with slight progresses. This is especially evident from the fragmentation of current bicycle network that intermittently connects with main trunk lines of Bangkok's transportation systems [61]. Nonetheless, we expect that the spatio-temporal characteristics of Pun Pun, revealed by this paper, would be found useful in the development of more sustainable bicycling infrastructure, as they are derived from real patron's usage rather than the projection from the long-gone events.

## 2.2. Bicycle-sharing system in Bangkok

Pun Pun is a public bicycle-sharing system (with docking stations) operated within the Central Business District (CBD) of Bangkok, where high-rise and high-density development has taken place. The system was initially built to encourage active transportation and to serve as a first- and last-mile solution for the scattered CBD's metro stations in 2012, with only 12 stations [62]. Afterwards, the number of stations gradually increased, together with significant daily trip growth, which became more stable in 2014. Currently, Pun Pun has about 50 bicycle stations across the CBD of Bangkok [63], each of which has exactly 8 slots for parking bicycles, as illustrated in Fig 1.

From Fig 1, it could be seen that the current Pun Pun's stations are rather dense within the center of Bangkok–many of which are located along the main roads in Chong Nonsi-Sathorn district or located close to main metro/subway stations, next to Lumpini park and Chulalongkorn University. Besides these areas, the bicycle-sharing stations are comparatively sparse, with limited coverage, making Pun Pun a less convenient means of transportation in parts of the CBD.

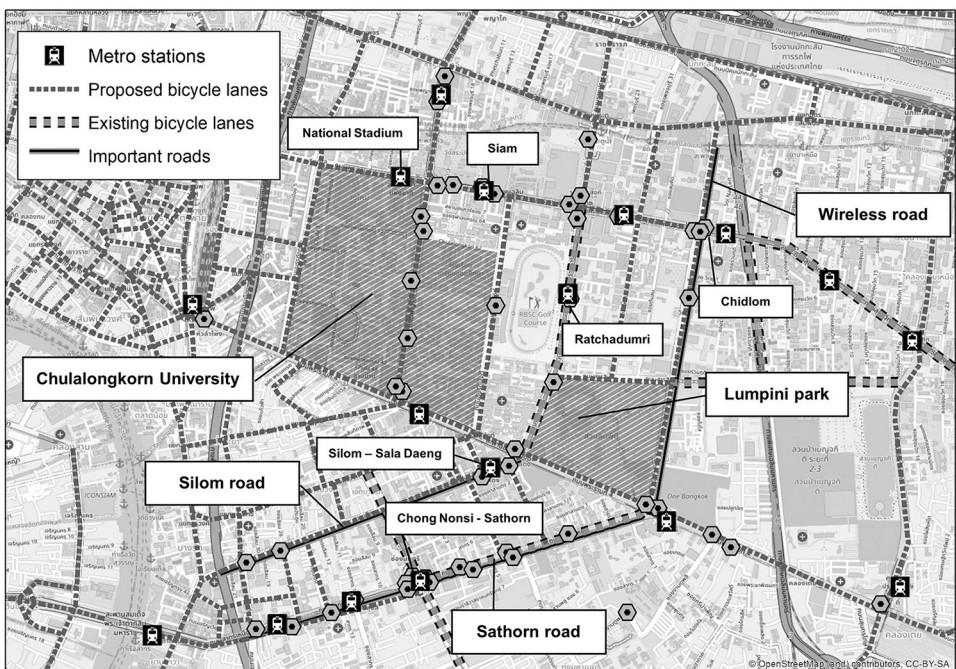

**Fig 1. Locations of Pun Pun's stations and important places in their vicinity.** The figure contains a base map from OpenStreetMap and OpenStreetMap Foundation, which is made available under the Open Database License.

It should be remarked that, apart from transportation, Pun Pun also serves as a prominent city attraction connecting the whole CBD with various modes of transportation (*e.g.* metros, buses, *Tuk Tuk*s, and motorcycle taxis). Besides, it is the only remaining public bicycle-sharing system in Bangkok, due to the closure of other public bicycle-sharing operators, such as Mobike and Ofo, in 2019 [64,65].

### 2.3. The first pandemic wave in Bangkok

The first COVID-19 case in Thailand was confirmed on January 12, 2020; and, from then on, the number of cases gradually rose, mostly in the BMR, due to its dense population. In order to keep the pandemic under control, the Thai government first issued a National Emergency Order (NEO) on March 25, 2020, and later imposed a nationwide curfew between 10 PM and 4 AM on April 3, 2020. Following these orders, all international and domestic flights were suspended, along with the closure of non-essential businesses, while all of the operating public transportation systems must place limits on the numbers of passengers to avoid crowdedness. These stringent countermeasures seemed to be greatly effective as the number of locally transmitted cases gradually declined, with no locally transmitted cases reported since May 23, 2020, as illustrated in Fig 2.

Although the number of COVID-19 cases begins to rise again in December 2020, this study will focus only on impacts of the first pandemic wave on Pun Pun, mainly because of data limitation and the fact that the current pandemic wave has not yet come to its end. While we expect that the mobility patterns of this transportation mode, as well as those of the others, might further change as the pandemic prolongs, our proposed tensor decomposition-based framework is still applicable, as we may include new datasets and run the experiments once again, with slight modifications.

### 3. Data collection

This section briefly describes the data collection process, where two different datasets are gathered for the analysis of Pun Pun, including (*i*) the data of bicycle-sharing trips and (*ii*) the spatial data of bicycle-sharing stations, along with those of the Bangkok CBD's street network. In addition to these datasets, trip data of other transportation modes in Bangkok during the pandemic are also collected for a comparative study.

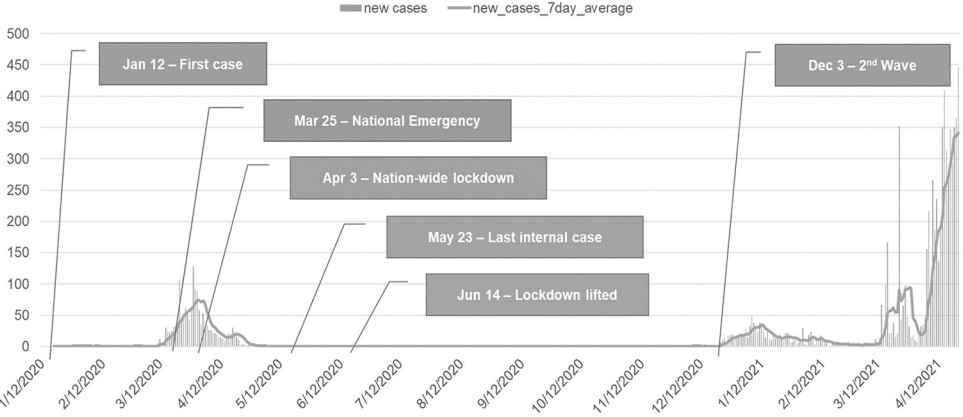

**Fig 2. COVID-19 cases in Bangkok and the government measures.**

## 3.1. Trip data

The data of Pun Pun's trips are provided by the BMA's Traffic and Transport Department, as well as its operator, namely the Smart Bike Service (SBS), whose coverage includes over 103,000 bicycling trips between 2018 and 2020 –parts of this dataset are, however, missing from April 17 to May 3, 2018, and from December 16, 2018, to February 23, 2019, due to server's technical issues. For each trip record, the data entail user's and bicycle's identification numbers, locations of rental and return (stations), and its respective rental time.

Trip data of other transportation modes, on the other hand, are obtained from the Thailand's Open Government Data [66]. This dataset contains the information of daily transit use in three Bangkok's public transportation systems, namely fixed-route buses, rails, and ferries, as well as that of personal trips on highways; but, it does not include trip data of demand-responsive transport such as taxis, motorcycle taxis, and modified pickup trucks (typically referred to as *Song Thaeow* and *Silor Lek*).

## 3.2. Bicycle-sharing stations and the Bangkok CBD's street network

The locations of all Pun Pun's stations are also extracted from the Open Government Data of Thailand, which are later embedded into the Bangkok CBD's street network by OpenStreet-Map. In addition to these bicycle-sharing stations, six more groups of locations are included for the spatial analysis of Pun Pun's trips (as summarized in Table 1), which are herein referred to as the points of interest (POIs). Once pinned, the distance and time to traverse among these locations are then calculated by OSMnx, a Python package used for analyzing real-world street networks and geospatial geometries [67].

# 4. Methodology

## 4.1. Tucker decomposition

In order to analyze the multi-dimensional characteristics of Pun Pun, especially its spatial and temporal dimensions in pre- and post-pandemic periods, a variant of tensor-based approaches–namely, a non-negative Tucker decomposition approach–has been herein adapted and applied to the acquired datasets. Similar to other traditional decomposition approaches,

**Table 1. Categories of POIs.**

| Category | Sub-category (Open Street Map) |
|---|---|
| **Transportation (TRA)** | railway station, bus stop, taxi, ferry terminal, bus station, helipad |
| **Commercial (COM)** | track, supermarket, pitch, mall, bank, hotel, department store, nightclub, restaurant, car dealership, beauty shop, convenience, food court, hairdresser, optician, chemist, hostel, guesthouse, post office, jeweler, bar, fast food, biergarten, cafe, golf course, pub, clothes, tourist info, doctors, telephone, atm, pharmacy, post box, bookshop, veterinary, kiosk, bicycle shop, furniture shop, do-it-yourself, toy shop, motel, shoe shop, bakery, car wash, travel agent, computer shop, florist, laundry, dentist, bench, drinking water, stationery, theatre, butcher, gift shop, sports shop, beverages, newsagent, car rental, mobile phone shop, greengrocer, outdoor shop, vending any, tower, public building |
| **Leisure (LEI)** | park, sports center, stadium, attraction, graveyard, castle, swimming pool, arts center, fountain, cinema, zoo, museum, wayside shrine, memorial, monument, playground, artwork, viewpoint, ice rink, picnic site |
| **Government (GOV)** | Police, hospital, embassy, courthouse, wastewater plant, community center, water tower, town hall, recycling paper, fire station |
| **Education (EDU)** | School, university, library, college, kindergarten |
| **Residential (RES)** | Shelter, residential |

such as matrix singular value decomposition or principal component analysis, Tucker decomposition could help extract patterns of complex datasets, as well as the changes and inter-relationships among different data dimensions, but with less computational resource. Tucker decomposition is also found superior for the analysis of higher-dimensional data vectors–typically referred to as tensors–especially the extraction of spatio-temporal patterns of trip data [68], as it allows users to analyze such vectors taking into consideration interactions among different data dimensions concurrently [69]. More importantly, the patterns of trips identified by Tucker decomposition tend to be more accurate and easy to interpret [68]. Since Tucker decomposition could be further generalized for higher-dimensional data analyses–which is, sometimes, referred to as Higher Order Singular Value Decomposition (HOSVD)–Tucker decomposition has been, therefore, regarded as one of prominent tools that has been widely applied in the study of complex trip patterns [19,21,70–72].

Technically speaking, Tucker decomposition is a technique that decomposes a tensor into a small core tensor multiplied with a corresponding factor matrix in each data dimension, or mode. To this end, a core tensor could be regarded as an information array indicating the degree of interactions among modes, while each factor matrix provides specific information pertaining to such a mode. Fig 3, for instance, illustrates an example of the third-order Tucker decomposition, where a tensor $X \in R^{N_A \times N_B \times N_C}$ is decomposed into a core tensor $Y \in R^{K_A \times K_B \times K_C}$ and three factor matrices $A \in R^{N_A \times K_A}$, $B \in R^{N_B \times K_B}$, and $C \in R^{N_C \times K_C}$ by Eq 1.

$$X \approx \hat{X} = Y \times_A A \times_B B \times_C C = \sum_{i=1}^{K_A} \sum_{j=1}^{K_B} \sum_{k=1}^{K_C} y_{i,j,k} a_i \odot b_j \odot c_k, \tag{1}$$

where $\hat{X}$ denotes the approximate tensor recovered by the abovementioned decomposition; $y_{i,j,k}$ is an element of the core tensor associated with the $i^{th}$, $j^{th}$, and $k^{th}$ columns of factor matrices $A$, $B$, and $C$; $K_A$, $K_B$, and $K_C$ denote ranks (patterns) of factor matrices $A$, $B$, and $C$; $\times_n$ is the $n$-mode matrix product of a tensor; and $a_i$, $b_j$ and $c_k$ denote the $i^{th}$, $j^{th}$, and $k^{th}$ column of factor matrices $A$, $B$, and $C$, respectively.

Following Fig 3 and Eq 1, as a tensor $X$ is represented by an approximate tensor $\hat{X}$, we may therefore formulate Tucker decomposition as an optimization problem by Eq 2.

$$F = \frac{1}{2} \|X - Y \times_A A \times_B B \times_C C\|^2, \tag{2}$$

where $\|\cdot\|^2$ denotes the Frobenius norm (similar expressions could be constructed for higher-

## Ex. Third-order tensor

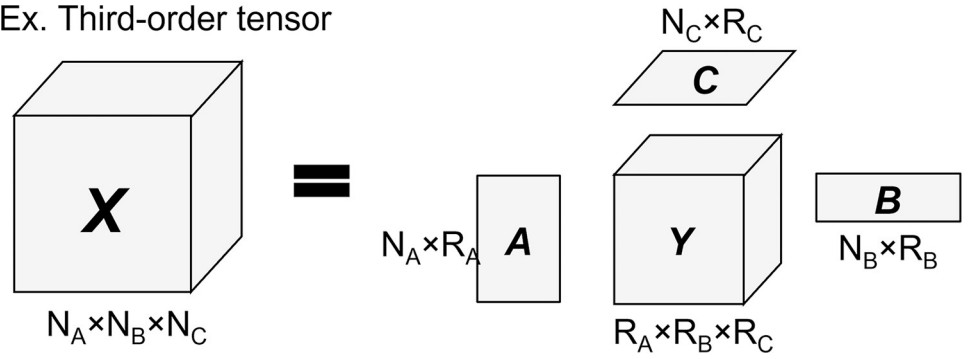

**Fig 3. Example of the third-order Tucker decomposition.**

dimensional vector decomposition, *e.g.* the fourth-order tensor decomposition, as in [73] and this present work).

From Eq 2, it could be seen that Tucker decomposition could potentially produce non-positive elements in the core and factor matrices–which are difficult to interpret, especially in the real-world applications [74,75]. A non-negative Tucker decomposition approach has been therefore devised to avoid the extraction of non-positive trip data. The concept of non-negative Tucker decomposition is quite simple as we add non-negativity constraints to Eq 2 to prevent *Y*, *A*, *B*, and *C* from taking negative values [76,77]; and, this could be performed via the opensource Python package, called TensorLy [78].

It is worth remarking that, prior to the factorizing process, the number of ranks for each mode–*e.g.* $K_A$, $K_B$, and $K_C$–must be predetermined to better represent the detailed characteristics (or patterns) of such a mode in the decomposition. Although higher number of ranks usually improves the quality of decomposition, it often complicates the interpretation of decomposition results [74]. As such, one must carefully determine an appropriate number of ranks for each mode before applying Tucker decomposition–which could be done via subjective analyses with some measurement metrics [68,75]. To this end, the Kullback–Leibler (KL) divergence is among the metrics that has been widely used in the evaluation of difference between two matrices, where the closer the two tensors, the lower the value of KL divergence [68,79]. Mathematically, the KL divergence is derived from a scoring technique, whose expression is shown by Eq 3 below.

$$D_{KL}(X||\hat{X}) = \sum_{i=1}^{N_O} \sum_{j=1}^{N_D} \sum_{k=1}^{N_{D_N}} \sum_{l=1}^{N_{H_N}} x_{i,j,k,l} log \frac{x_{i,j,k,l}}{\hat{x}_{i,j,k,l}} - x_{i,j,k,l} + \hat{x}_{i,j,k,l};\ x_{i,j,k,l} \in X, \hat{x}_{i,j,k,l} \in \hat{X}. \quad (3)$$

## 4.2. Tensor of Pun Pun's dataset

Since each bicycle-sharing trip records the use of a bicycle between a pair of stations, *i.e.* an origin-destination pair (OD pair), on a particular date at a particular time period, the tensor of Pun Pun could be therefore constructed as a fourth-order tensor, whose size is [$O$ \* $D$ \* $D_N$ \* $H_N$], where (*i*) $O$ denotes the number of origins ($O \in R^{N_O \times K_O}$, $N_O = 50$ stations), (*ii*) $D$ denotes the number of destinations ($D \in R^{N_D \times K_D}$, $N_D = 50$ stations), (*iii*) $D_N$ denotes total number of days in the study, excluding days with missing information ($D_N \in R^{N_{D_N} \times K_{D_N}}$, $N_{D_N} = 1,010$ days), and (*iv*) $H_N$ denotes the time period (*i.e.* the operating hours between 5 AM– 8 PM) at which a trip starts ($H_N \in R^{N_{H_N} \times K_{H_N}}$, $N_{H_N} = 14$ time periods).

For each mode, the number of ranks (*i.e.* $K_O$, $K_D$, $K_{D_N}$, and $K_{H_N}$) is determined based on the resulting non-negative approximate tensor ($\hat{X}$) that provides the smallest KL-divergence value. Regarding the rank of time-of-day mode, we find that bicycling demand consistently has two peak times on weekdays, and it is roughly constant on weekends, leading to an estimated rank of three or four for this temporal dimension. However, as there is no evident pattern for the remaining information dimensions, the numbers of ranks for these modes are then set within predefined ranges: three to eight for day mode and two to nine for both origin and destination modes (spatial dimensions). Based on this setting, we find that, among 96 combinations, the optimal rank of core tensor that provides the least KL-divergence value defined by Eq 3 is: $K_O$ = 7, $K_D$ = 7, $K_{D_N}$ = 4, and $K_{H_N}$ = 3 (see Table 2 for more detailed KL-divergence results). Similar to [68,74], we find that KL-divergence values tend to decrease with higher rank values; however, after the optimal ranks are determined, KL-divergence values marginally change.

**Table 2.  KL-divergence results.**

| mode (O, D) | | mode (Day) | | | | | |
|---|---|---|---|---|---|---|---|
| | | **3** | **4** | **5** | **6** | **7** | **8** |
| **2** | | 493,091.44 | 498,212.19 | 499,221.65 | 965,110.19 | 808,044.69 | 501,624.30 |
| **3** | | 517,872.61 | 508,981.43 | 609,229.45 | 628,402.83 | 713,613.44 | 623,373.12 |
| **4** | | 483,775.70 | 535,350.43 | 532,892.81 | 590,285.57 | 591,049.74 | 576,273.12 |
| **5** | | 538,595.58 | 531,499.55 | 534,683.66 | 543,360.77 | 559,209.14 | 562,766.40 |
| **6** | | 463,766.31 | 439,566.11 | 453,978.75 | 502,178.32 | 504,246.39 | 490,180.02 |
| **7** | | 438,092.44 | **436,638.18***  | 438,936.76 | 498,771.16 | 440,317.92 | 467,200.39 |
| **8** | | 445,681.28 | 450,573.89 | 454,722.53 | 477,124.69 | 476,429.74 | 474,825.74 |
| **9** | | 445,970.71 | 443,796.18 | 458,716.68 | 466,225.52 | 469,554.13 | 468,287.00 |

* The lowest value of KL divergence.

## 5. Results and discussions

### 5.1. Descriptive analysis

According to our datasets (*i.e.* trip data between 2018–2020), it is evident that the number of bicycle-sharing trips is largely affected by the first pandemic wave, especially when the number of confirmed cases is rising, as illustrated by Fig 4. To be precise, the number of trips is rather consistent, with a slight decrease during the summer months (March–May), until the end of 2019, when the global pandemic begins. Since then, the number of trips starts decreasing and later drops sharply as the number of COVID-19 cases surges in March and April 2020. However, the number of trips seems to slowly recover after the end of the first pandemic wave, around the end of May 2020 –although it plummets once again at the end of 2020, as the second pandemic wave buffets Thailand.

A similar trend is also observed in the use of other transportation modes, in which the number of trips significantly drops during the pandemic due to restrictive traveling requirements and it then gradually recovers after such restrictions have been lifted (see Fig 5 for more details). Notwithstanding such a fact, the recovery rate of each system seems to be somewhat different. In particular, private transport has the quickest recovery rate followed by bicycle

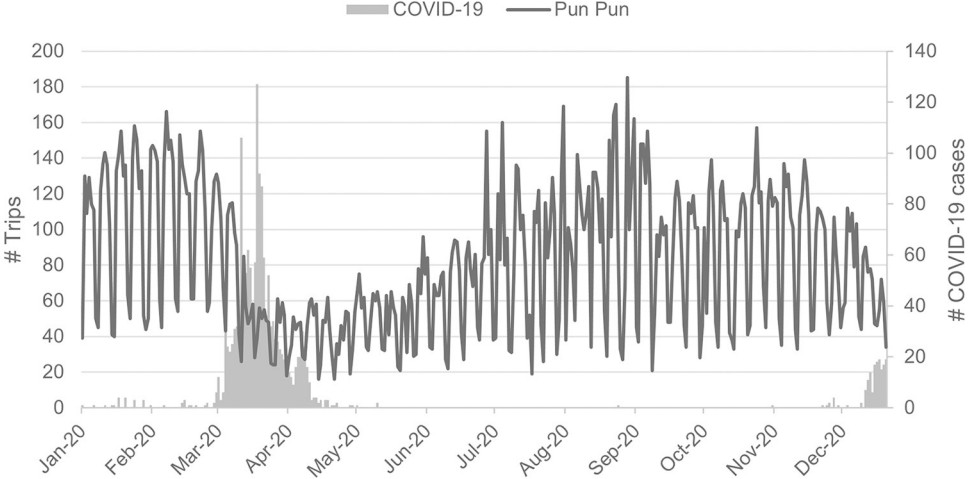

**Fig 4.  Bicycle-sharing trips and COVID-19 cases.** Daily bicycle-sharing trips and COVID-19 cases in 2020.

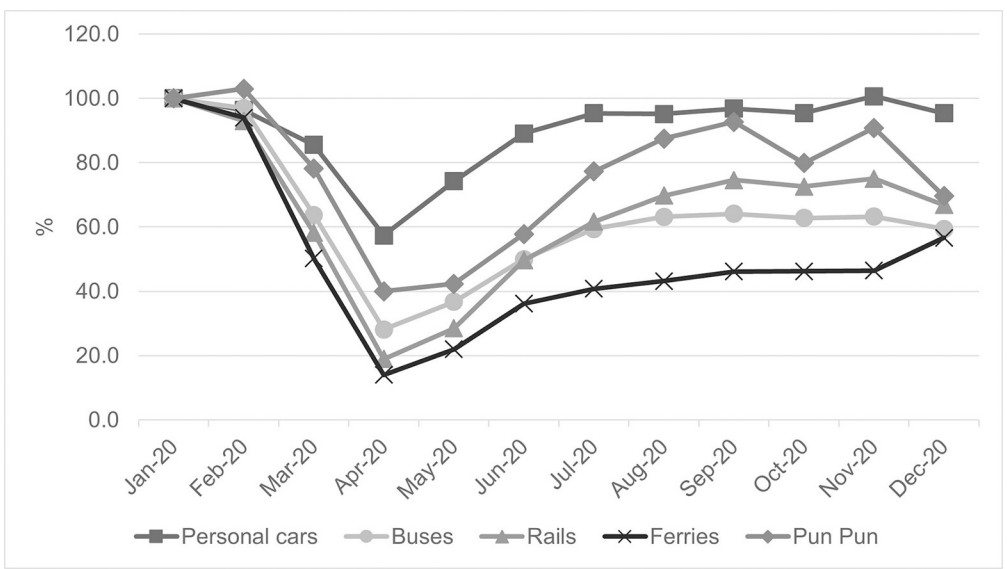

**Fig 5. Patterns of various transportation modes in 2020.**

sharing, while other public transportation systems seem to suffer badly from the pandemic with relatively slow recovery rates, presumably because of changes in transport behavior mentioned earlier.

Regarding trip patterns, we find that average weekday trips of Pun Pun are almost triple of those during weekends. Moreover, while there are clearly two peaks on weekdays, namely, the morning peak from 6 AM to 9 AM and the evening peak from 4 PM to 7 PM, the number of trips during weekends is relatively more stable throughout the days, as shown in Fig 6. Similar trip patterns have also been detected in pre- and post-pandemic periods, but with different average numbers of trips, confirming the existence of morning and evening peaks during weekdays and that of off-peak during weekends.

In terms of trip durations, it could be seen from Fig 7 that the median trip duration of Pun Pun is increasing from 9.38 minutes in 2018 to 14.2 minutes in 2020, while percentage of journeys that begin and end at the same stations is also rising from 15% in 2018 to 27% in 2020. These findings suggest that bicycle sharing receives more attention from Bangkok residents during the pandemic, which is in-line with the works by [21,80,81].

## 5.2. Spatio-temporal analysis of Pun Pun

In order to visualize the impacts of COVID-19 pandemic on Pun Pun, we first provide analyses related to the spatial and temporal dimensions of Pun Pun, in Sections 5.2.1 and 5.2.2. Then, the interactions between these two dimensions, especially in pre- and post-pandemic periods, are thoroughly discussed in Section 5.2.3. Finally, managerial implications are suggested in Section 5.3.

**5.2.1. Spatial characteristics of Pun Pun.** According to the optimal ranks identified by the KL divergence, bicycle-sharing stations could be divided into seven groups–each of which represents a unique spatial characteristic of bicycle-sharing trip's patterns, *i.e.* how frequently people begin and end their rides at each station–for both origin and destination modes, as shown in Figs 8 and 9 (darker dots in each group denote stations with relatively more usage frequency–or, equivalently the hotspots in the literature [68]). To better describe these spatial modes, the intensity of nearby POIs to each station (within 500 m) has been included as

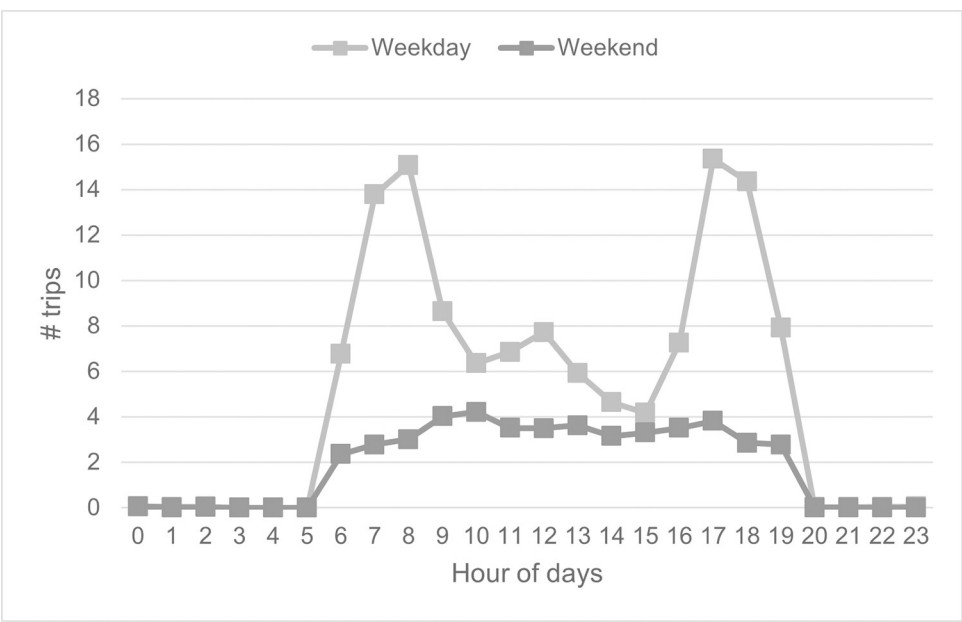

**Fig 6. Average number of trips on weekdays and weekends.**

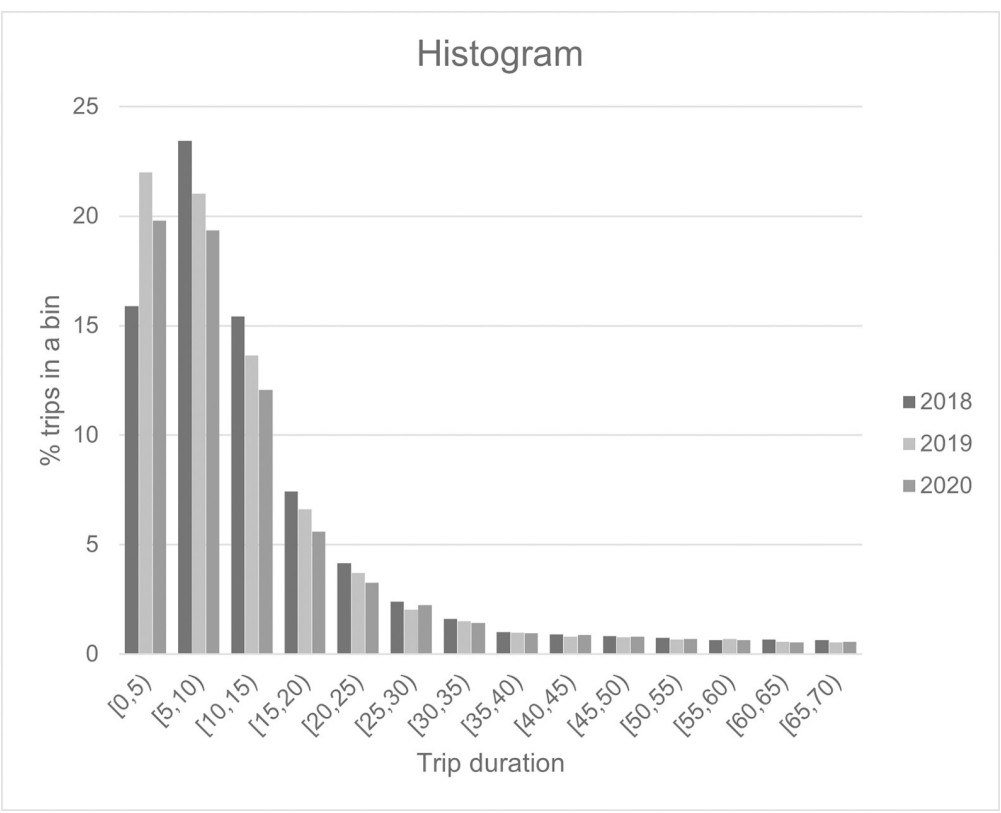

**Fig 7. Histogram of trip durations.**

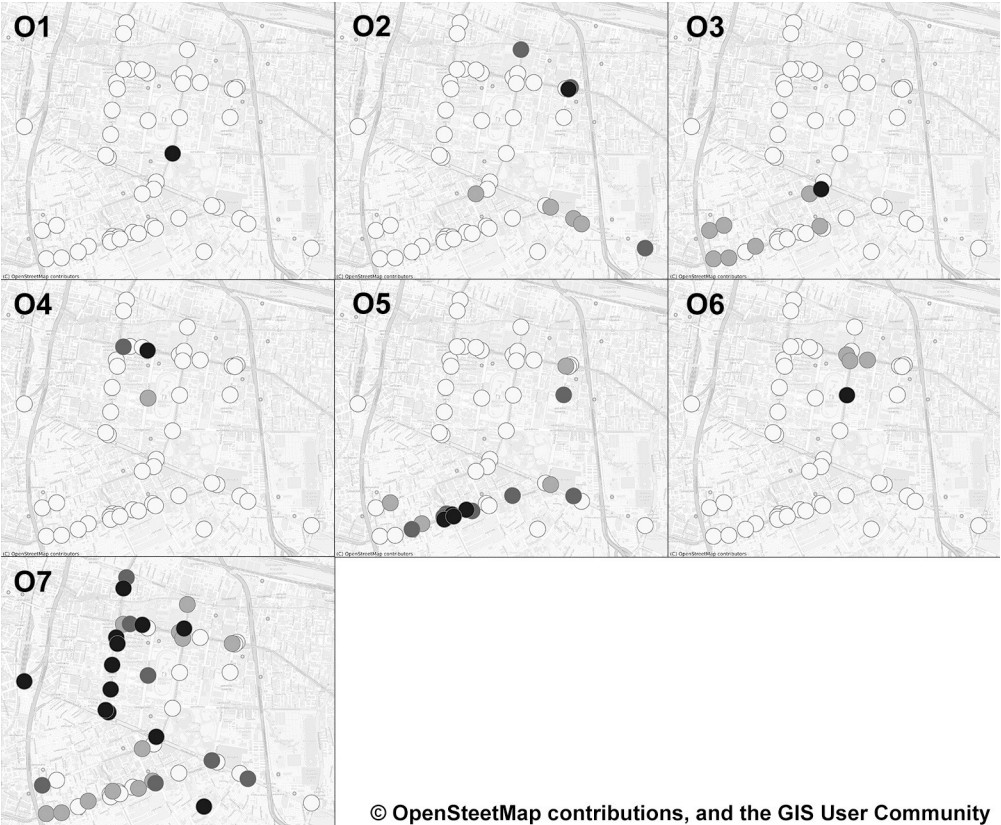

**Fig 8. Groups of origin stations identified by a non-negative Tucker decomposition approach.** These figures contain a base map from OpenStreetMap and OpenStreetMap Foundation, which is made available under the Open Database License.

shown in Fig 10 (please see Table 1 and Fig 1 for the detailed information of POIs and locations of important places within the Bangkok's CBD).

Based on these figures, Lumpini 1 station (*i.e.* the darkest dot in sub-figures O1 and D1) is regarded as the main bicycle-sharing station in O1 and D1, due largely to its location that connects nearby metro stations with Lumpini Park, the CBD's largest public park. Unlike O1 and D1, the most frequently used stations in O2 and D2 lie within the dense mixed-use, high-rise, and luxurious buildings of Chidlom district and Wireless road, while those of O3 are placed near Silom-Sala Daeng subway station surrounded by nightclubs, luxury condominiums, and commercial offices. However, such stations are interestingly not a part of any specific destination modes.

Similar to O3, the most frequently used stations in O4 and D4 are located around Siam-National Stadium metro stations, but those of D4 are more dispersed towards Chulalongkorn University. Likewise, the most frequently used stations of O6 and D5 are found within the vicinity of Ratchadamri metro station, while those of O5, D3, and D6 are quite similar as they are clustered around Chong Nonsi and Sathorn metro stations–one of Bangkok's primary business and commercial districts. Lastly, the characteristics of O7 and D7 are relatively more dispersed and scattered throughout the Bangkok's CBD.

**5.2.2. Temporal characteristics of Pun Pun.** Regarding the time-of-day mode, a non-negative Tucker decomposition approach has revealed three distinct patterns, including (*i*) morning-peak, (*ii*) evening-peak, and (*iii*) off-peak as illustrated by Fig 11. For the morning peak, labeled as AM peak mode, the values of a factor matrix are higher between 6 AM and 9

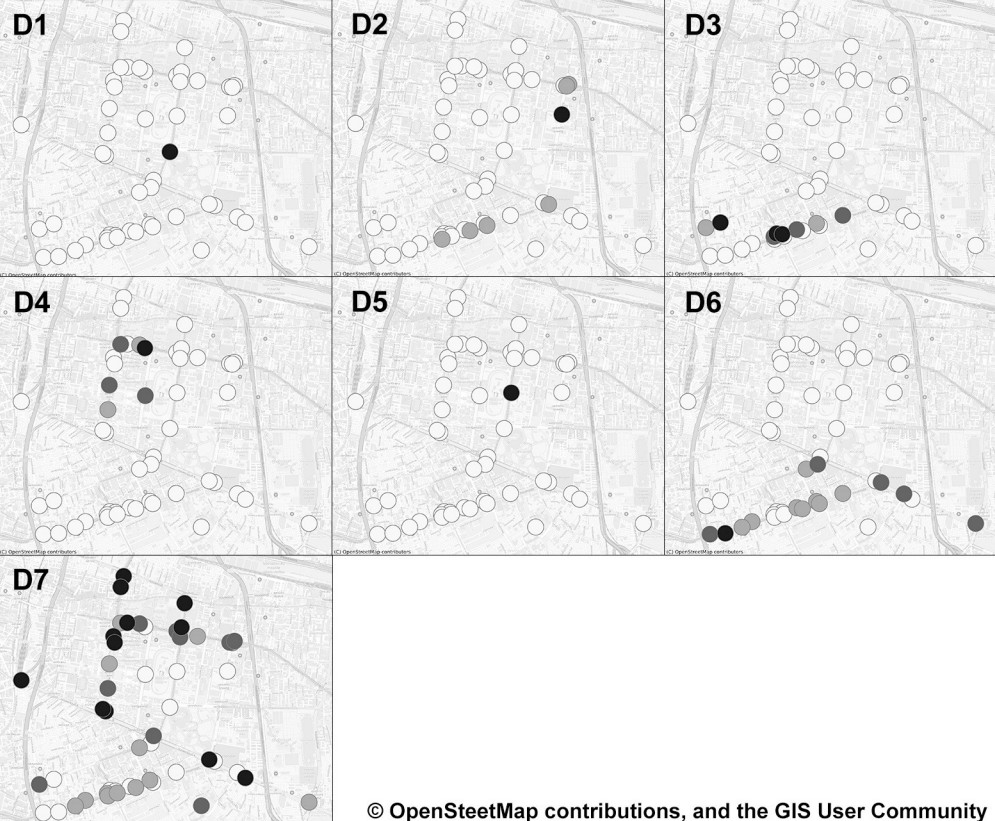

© OpenSteetMap contributions, and the GIS User Community

**Fig 9. Groups of destination stations identified by a non-negative Tucker decomposition approach.** These figures contain a base map from OpenStreetMap and OpenStreetMap Foundation, which is made available under the Open Database License.

AM, while those of the evening peak, labeled as PM peak mode, are higher between 4 PM and 7 PM. These values confirm both peaks of bicycle-sharing trips in Fig 6; and, they also indicate that patrons tend to ride a shared bicycle for business trips. Unlike morning and evening peaks, the off-peak mode clearly has a spike around 10 AM, resulting from weekend trips with a slightly higher number of records during such a period.

Regarding the day mode, a non-negative Tucker decomposition approach has detected four different trip patterns that dynamically change over the period of study, on both weekdays and weekends, denoted by T1 –T4 in Fig 12(A) and 12(B), respectively. For ease of discussion, monthly mean factor values are reported rather than daily factor values, where trip patterns with higher monthly mean factor values are typically those that dominate others in terms of trip numbers in a particular month. Besides, the shaded areas in these figures represent the periods during which Pun Pun encountered technical difficulties, which results in loss of data.

Based on Fig 12, it could be seen that, prior to the pandemic, the tendency of trips, excluding T1, likely drops during the summer months, due to the extremely hot weather. And, most trip patterns are adversely affected by the pandemic, especially those on weekdays, as their monthly mean factor values greatly decline. Furthermore, transport behavior of Pun Pun's users seems to also change after the first pandemic wave ends, which is evident from the rise of T2 trips that dominate other trip patterns on both weekdays and weekends.

**5.2.3. Spatio-temporal characteristics of Pun Pun.** To better understand spatio-temporal characteristics of Pun Pun, all major origin-destination pairs have been further attached to

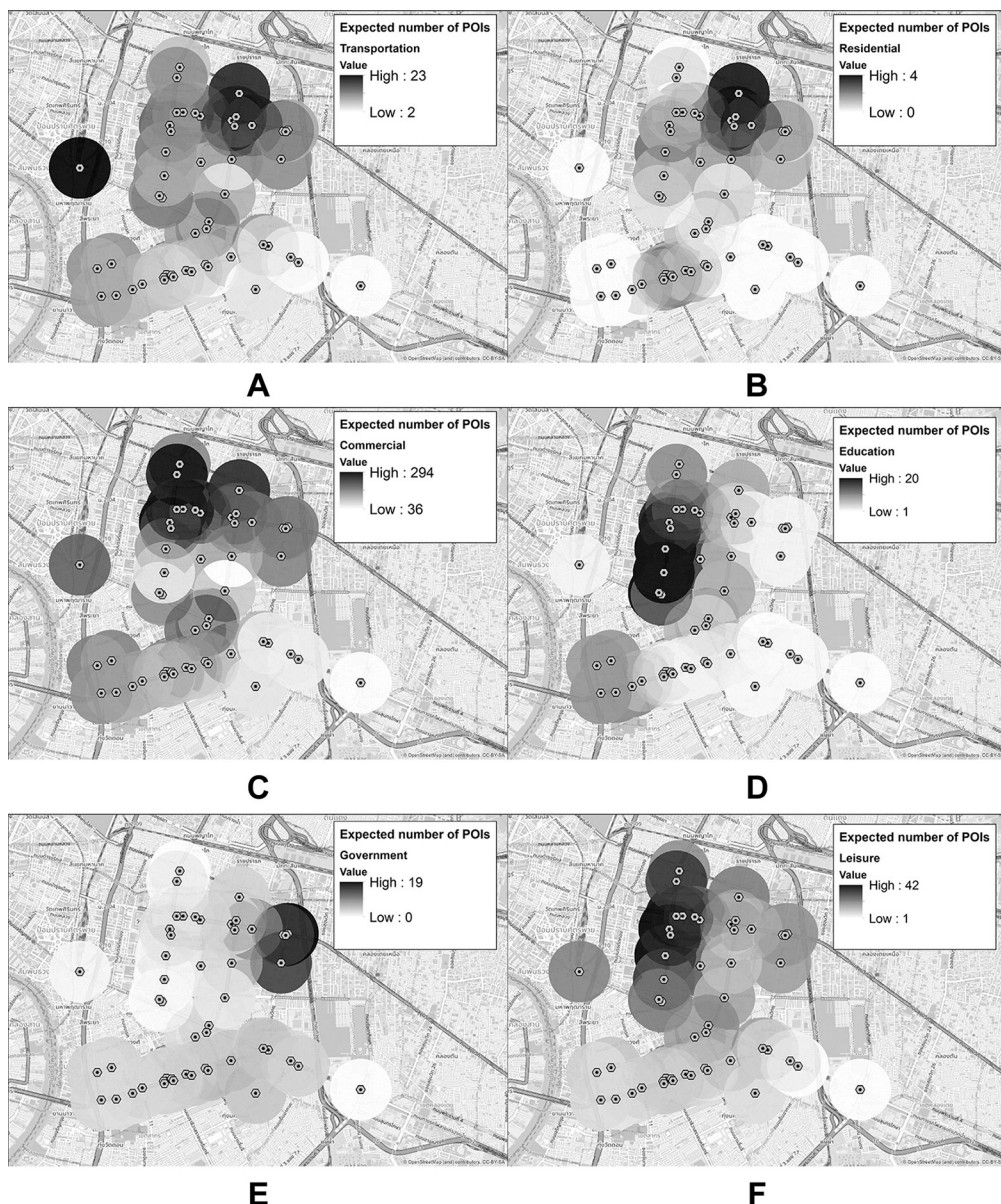

**Fig 10. Intensity of each POI within 500 m from Pun Pun's stations.** (A) Transportation. (B) Residential. (C) Commercial. (D) Education. (E) Government. (F) Leisure. These figures contain a base map and data from OpenStreetMap and OpenStreetMap Foundation, which is made available under the Open Database License.

all three peak periods of all trip patterns, as reported in Table 3. From Table 3, it has become much clearer that Pun Pun's trips are likely to originate from or destine to a public park or commercial districts with close connections to Bangkok's metro/subway stations. T1 trips made during the morning peak, for instance, typically involve bicycle-sharing stations within a congested area of Chidlom business district (O2/D2), while T3 trips made during the off-peak period (weekends), on the contrary, are likely recreational trips made by active users, as they both originate and destine within Lumpini park's areas. Also seen from Fig 12 that most trips made during the evening peak usually originate and destine along the main road that connects Chong Nonsi-Sathorn business district with nearby metro stations (O5/D6), which does make

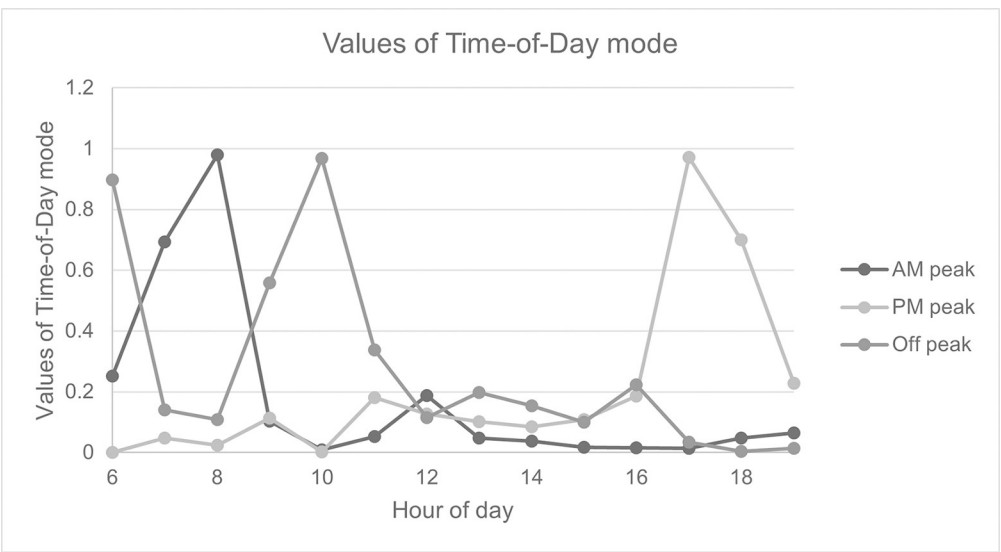

**Fig 11. Values of a factor matrix for time-of-day mode.**

a good sense for users who seek alternative first- and last-mile modes of transportation with less traffic congestion [82].

Regarding the impacts of COVID-19, we find that T1 and T4 trips made on weekdays are severely affected by the pandemic, even after the lift of all internal traveling restrictions. This is due to the fact that Chidlom (O2/D2), Ratchadumri (O6/D5), and Silom-Sathorn (O5/D6) districts are homes to a number of companies that still enforce their employees to work-from-home at that time, leading to an abrupt decrease in the number of trips, and so monthly mean factor values. Furthermore, majority of expatriate workers who live and work within these districts have already returned to their home countries before the complete lockdown. But, due to the worldwide international traveling restrictions, most are unable to return, causing a relatively slow recovery rate in these trips.

T2 trips, on the contrary, seem to be positively affected by the pandemic as they continue to rise, especially after the end of the first pandemic wave. Further investigations reveal that these trips normally involve stations that connect Siam-National Stadium metro stations with Chulalongkorn University (O4/D4) during the morning and off-peak periods, while connecting Chong Nonsi-Sathorn business district with nearby metro stations (O5/D6) during the evening peak. These findings may be regarded as a reflection of the ever-changing transport behavior of university students and people who seek an alternative first- and last-mile transportation mode in an urban area, with a more positive outlook on the use of bicycle-sharing system [83].

Unlike the abovementioned trip patterns, T3 trips are rather consistent during the period of study, especially in the off-peak period, with close connection to bicycle-sharing stations in the vicinity of Lumpini park (O1/O3/D1). Based on this finding, we may infer that T3 trips are made by active users who regularly use Pun Pun for recreational activities. Since bicycling requires less interaction among patrons–i.e. patrons can easily avoid exposure with others in a large open park area–T3 trips are therefore unlikely to be affected by the pandemic.

## 5.3. Managerial implications

According to identified trip patterns, most Pun Pun's users are possibly workers, including foreign nationals, who use Pun Pun to avoid traffic congestion in their daily life–as most

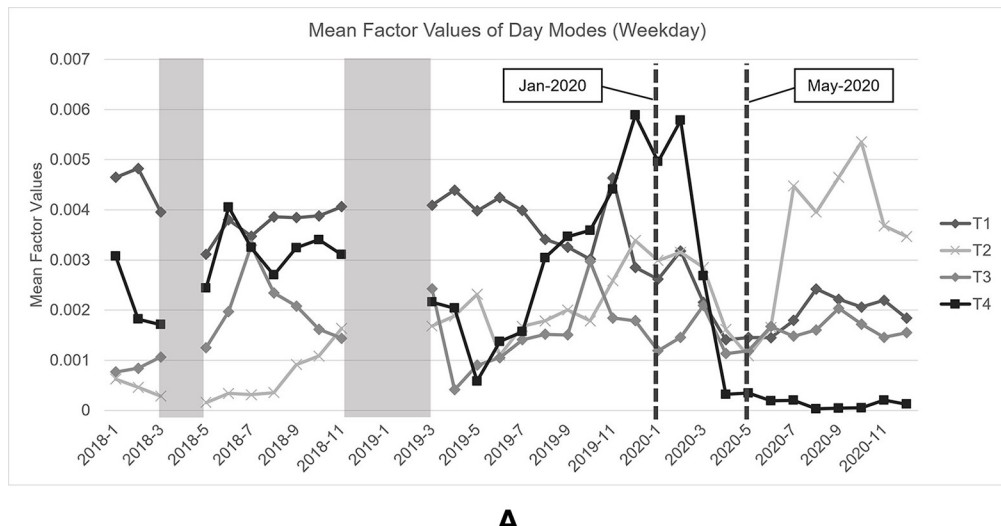

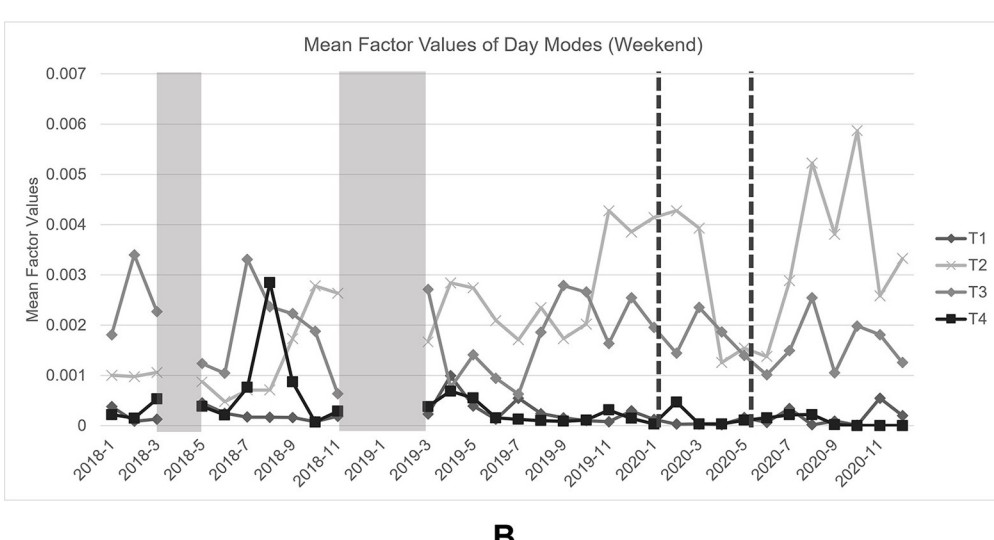

**Fig 12. Monthly mean factor values of day mode.** (A) Weekdays, and (B) weekends.

excursions begin and end in either residential or commercial districts, where multinational enterprises and organizations are concentrated. In addition to work-related bicycle-sharing journeys, another notable pattern occurs in public park areas, with close connections to some metro stations, indicating the role of Pun Pun as a potential first- and last-mile mode of transportation supporting limited trunk lines in such areas. These patterns are especially evident

**Table 3. Major origin-destination pairs of all trip patterns in three different peak periods.**

|    | AM peak | Off peak | PM peak |
|----|---------|----------|---------|
| **T1** | O2 / D2 | O6 / D5 | O5 / D6 |
| **T2** | O4 / D4 | O4 / D4 | O5 / D6 |
| **T3** | O3 / D3 | O1 / D1 | O5 / D6 |
| **T4** | O1 / D5 | O3 / D3 | O5 / D1 |

during the first pandemic wave, emphasizing a new perception of bicycle-sharing from a leisure activity to a more sustainable mode of green transportation.

Although Pun Pun is severely affected by the first pandemic wave, it is quite resilient when compared to other modes of public transportation, *i.e.* the number of Pun Pun's trips quickly returns to more than 77% of the original level, soon after the first pandemic wave ends. This finding is in-line with those of previous research conducting similar investigations, as in [51,83], confirming that bicycle-sharing is less vulnerable to external stimuli; and, commuters tend to look for a mode of transportation that best suits their purposes.

Despite positive impacts of bicycle sharing on urban transportation systems, the current benefits of Pun Pun are rather limited, due largely to Pun Pun's small operational scale, coupled with Bangkok's limited bicycling infrastructure (*e.g.* bicycle lanes). To better enhance the efficacy of Pun Pun–and so its contribution to the overall transportation system–Pun Pun's infrastructure should be further expanded in accordance with the ever-changing patron's behavior in a so-called new normal environment. In doing so, we have computed the shortest paths among all the trips and then suggested the bicycle network that supports all of the current trip patterns, as shown in Fig 13.

From Fig 13, it could be seen that the current bicycling infrastructure in the CBD of Bangkok is fairly ineffective, as it mainly supports trips made in Chong Nonsi-Sathorn district, Chulalongkorn University, and the main entrance of Lumpini park, with a number of missing connections among these three main areas. It is even worse since there are also some extended bicycle lanes to the South and the East of Bangkok, but there is, unfortunately, no Pun Pun's station to utilize such lanes. Although more bicycle lanes should be added to improve the

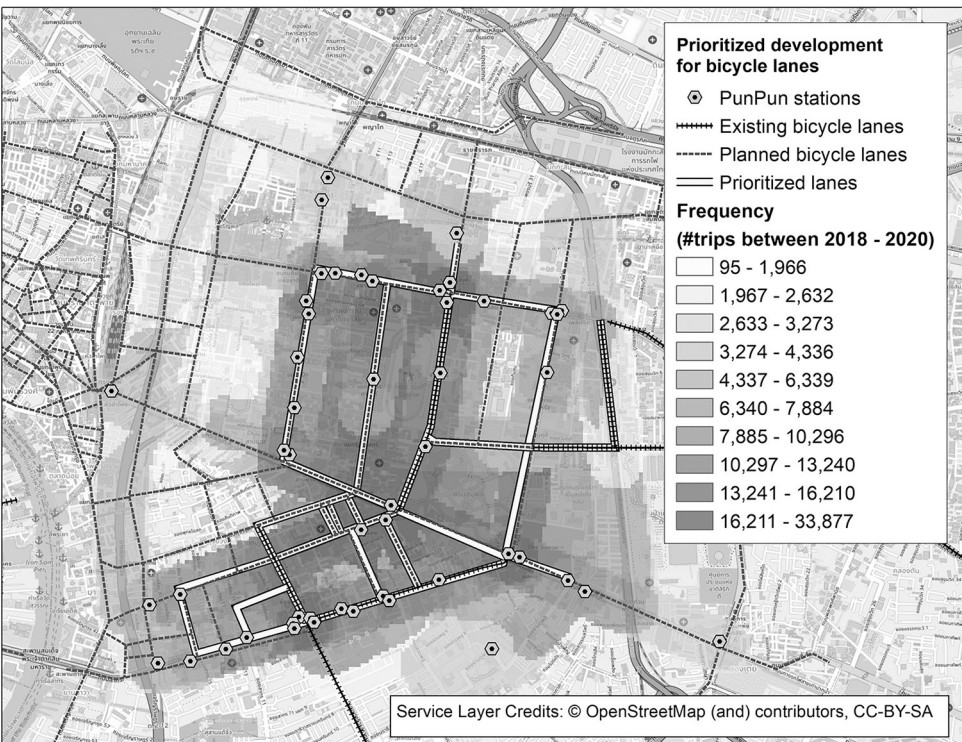

**Fig 13. Frequently used streets and a priority for a bicycling infrastructure development.** The figure contains a base map and data from OpenStreetMap and OpenStreetMap Foundation, which is made available under the Open Database License.

connectivity of this bicycle network, it is less likely that such an extension would move towards the BMA's plan, due to limited budgeting and changes in trip patterns that potentially make the investment less attractive. Rather, it is more sensible to focus on the construction of some crucial linkages that unify this fragmented network into a more connected one, as highlighted in Fig 13.

It should be remarked that, based on this proposed bicycle network, some more lanes should be constructed as planned, especially those connecting hotspots in the abovementioned three areas. However, there are also some streets that should be paid attention to, such as those providing shortcuts among main streets in Chong Nonsi-Sathorn district or those running parallel to the main streets for safety purposes. With greater connectivity, Pun Pun should better perform and serve as an alternative means of green first- and last-mile transportation as it is initially designed. We also expect that, with more trip data from various transportation systems, we would be able to extract more trip patterns and then combine them altogether for the construction of a more sustainable transportation network.

## 6. Conclusions and future work

COVID-19 is an ongoing pandemic that affects not only people lifestyle but also people behavior in various aspects, including transport mobility, which is evident from the shift of transportation modes towards private individual ones. Among these alternative modes of transportation, bicycle sharing has recently gained more public attention, due largely to its flexibility and social benefits that outweigh the others. Nonetheless, the success of bicycle sharing is rather limited, *i.e.* only in cities with sufficiently good bicycling infrastructure and legislation, while it is worse off in many other countries that regard bicycling as a merely means of leisure activities, including Bangkok, Thailand.

Considering these findings, this paper aims to explore Pun Pun, the only public bicycle-sharing system in Bangkok, Thailand, as well as the impacts of COVID-19 on such a system, based on the first pandemic wave in 2020. We also seek to determine whether Pun Pun and its patron's behavior have gradually moved towards the same directions as illustrated by the previous literature. In doing so, a non-negative Tucker decomposition approach has been adapted and applied to Pun Pun's datasets in order to extract patterns of Pun Pun's trips with respect to both spatial and temporal dimensions. We find that, while Pun Pun contributes only a small proportion to the daily trips made prior to the pandemic–accounting for approximately 30,000 trips annually–the outlook of such a system seems to be greatly improved after the first pandemic wave ends. The median trip duration, for instance, has been largely improved by 51.39%, while the percentage of journeys that begin and end at the same stations is also rising to 27% in 2020. More interestingly, we find that bicycle sharing is comparatively resilient, even in a city with relatively poor bicycling infrastructure like Bangkok, as the number of Pun Pun's trips gradually returns to more than 77% of the original level.

Regarding the spatio-temporal characteristics of Pun Pun, the non-negative Tucker decomposition has revealed four trip patterns–three of which are related to business trips with drastic changes in their patterns, while the remaining concerns recreational excursions that are less vulnerable to the external stimuli. Among the first three work-related trip patterns, there is one interesting trip pattern that benefits from the pandemic, namely the one that connects Chong Nonsi-Sathorn business district and Chulalongkorn University with nearby metro/subway stations. These findings infer that the adoption of Pun Pun is higher in groups of university students and people who seek an alternative mode of transportation with less traffic congestion in an urban area. They also shed light on the potentiality of Pun Pun as an alternative first- and last-mile mode of transportation complementing the current Bangkok's

transportation network–although a larger effort to the development of bicycling infrastructure should be further made, as discussed in Section 5.3.

Notwithstanding these findings, as the pandemic prolongs, the mobility patterns of this transportation mode, as well as those of the remaining, might be further changed. And, it would be interesting to see the course of such changes, along with the development of pandemic in the long run. We expect that the results to these extensions would be beneficial not only for policymakers in the planning of a more sustainable transportation network but also for the patrons who seek alternative modes of transportation that best suit their lifestyle–although more detailed information concerning user-specific behavior is in need. Lastly, it might be also interesting to include other private first- and last-mile transportation systems, such as CU Bike–the bicycle-sharing system provided by Chulalongkorn University in the Bangkok's CBD–into consideration, as there might be interactions among different first- and last-mile transportation systems within the same region. With this additional information, policymakers would be able to synthesize the existing systems and come up with policies or programs to promote all of these alternative transportation systems effectively and efficiently.

## Acknowledgments

We would like to express our gratitude to the Traffic and Transportation Department of Bangkok Metropolitan Administration (BMA) for the data of Pun Pun.

## Author Contributions

**Conceptualization:** Tawit Sangveraphunsiri, Tatsuya Fukushige.

**Formal analysis:** Tawit Sangveraphunsiri, Tatsuya Fukushige, Garavig Tanaksaranond, Pisit Jarumaneeroj.

**Funding acquisition:** Tawit Sangveraphunsiri.

**Investigation:** Tawit Sangveraphunsiri, Tatsuya Fukushige, Garavig Tanaksaranond, Pisit Jarumaneeroj.

**Methodology:** Tawit Sangveraphunsiri, Tatsuya Fukushige, Natchapon Jongwiriyanurak.

**Project administration:** Pisit Jarumaneeroj.

**Software:** Tawit Sangveraphunsiri, Tatsuya Fukushige.

**Supervision:** Garavig Tanaksaranond, Pisit Jarumaneeroj.

**Validation:** Garavig Tanaksaranond, Pisit Jarumaneeroj.

**Visualization:** Tawit Sangveraphunsiri, Tatsuya Fukushige.

**Writing – original draft:** Tawit Sangveraphunsiri, Tatsuya Fukushige, Natchapon Jongwiriyanurak.

**Writing – review & editing:** Garavig Tanaksaranond, Pisit Jarumaneeroj.

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
