## [Decision Letter · Decision Letter 0]

3 Jun 2022

PONE-D-22-13144Impacts of the COVID-19 pandemic on the spatio-temporal characteristics of a bicycle-sharing system: A case study of Pun Pun, Bangkok, Thailand PLOS ONE

Dear Dr. Jarumaneeroj,

Thank you for submitting your manuscript to PLOS ONE. After careful consideration, we feel that it has merit but does not fully meet PLOS ONE’s publication criteria as it currently stands. Therefore, we invite you to submit a revised version of the manuscript that addresses the points raised during the review process.

We look forward to receiving your revised manuscript.

Kind regards,

Sathishkumar V E

Academic Editor

PLOS ONE

Journal Requirements:

"This paper is supported by the Second Century Fund (C2F), Chulalongkorn University, Bangkok, Thailand. We would like to express our gratitude to the Traffic and Transportation Department of Bangkok Metropolitan Administration (BMA) for the data of Pun Pun."

"TS is supported by the Second Century Fund (C2F), Chulalongkorn University, Bangkok, Thailand (https://www.research.chula.ac.th/the-second-century-fund-chulalongkorn-university-c2f). The funder had no role in study design, data collection and analysis, decision to publish, or preparation of the manuscript."

4. We note that Figures 1,8,9 and 10 in your submission contain [map/satellite] images which may be copyrighted. All PLOS content is published under the Creative Commons Attribution License (CC BY 4.0), which means that the manuscript, images, and Supporting Information files will be freely available online, and any third party is permitted to access, download, copy, distribute, and use these materials in any way, even commercially, with proper attribution. For these reasons, we cannot publish previously copyrighted maps or satellite images created using proprietary data, such as Google software (Google Maps, Street View, and Earth). For more information, see our copyright guidelines: http://journals.plos.org/plosone/s/licenses-and-copyright.

a. You may seek permission from the original copyright holder of Figures 1,8,9 and 10 to publish the content specifically under the CC BY 4.0 license.  

Reviewers' comments:

Reviewer's Responses to Questions

**Comments to the Author**

1. Is the manuscript technically sound, and do the data support the conclusions?

Reviewer #1: No

Reviewer #2: No

2. Has the statistical analysis been performed appropriately and rigorously? 

Reviewer #1: No

Reviewer #2: Yes

3. Have the authors made all data underlying the findings in their manuscript fully available?

Reviewer #1: No

Reviewer #2: No

4. Is the manuscript presented in an intelligible fashion and written in standard English?

Reviewer #1: No

Reviewer #2: Yes

5. Review Comments to the Author

Reviewer #1: 1. Structure of the paper is not well defined. Paper represents very basic work with very less scientific proofs.

2.Motivation of the proposed scheme is not clear.

3.Mathematical analysis of the proposed scheme is very weak.

4. Results and analysis section should be further improved.

5.The paper also contains a lot of typos that must be revised.

Include & cite recent publications from current year

6.Abstract Section - Authors are suggested to include their best results of the proposed method to give an overall picture and highlights their contribution towards the field of study.

7.Authors are suggested to include more discussion on the results and also include some explanation regarding the justification to support why the proposed method is better in comparison towards other methods.

8.Conclusion Section - Authors are suggested to highlight their exact best results in comparison of other methods to justify the advantages of their proposed method.

Reviewer #2: This paper proposes a tensor-based framework to reveal the changes in transportation modes influenced by COVID-19. It provides a relatively comprehensive analysis of the spatio-temporal patterns of the bike-sharing system during the first wave of the pandemic. This paper adopted various analysis and visualization approaches, but in my opinion, significant improvement should be made before this manuscript is accepted.

First, this manuscript should emphasize why the topics concerning bike-sharing and the findings of your research are crucial. This manuscript tries to analyze the daily patterns of bike-sharing, variations in the number of trips made by bike-sharing and other transportation modes, and spatial patterns of OD. But these results are not tightly connected with each other to serve a better understanding of the motivation of the research. What is the scientific contribution of the research? The research should clarify the innovation/purpose of your research and reorganize the manuscript.

Second, the paper proposes a tensor-based framework for analyzing the impacts of the COVID-19 pandemic on Pun Pun. The paper should demonstrate why this framework is an efficient approach. Regarding the findings considering the temporal and spatial patterns, I am not fully convinced as the findings – finding the stations with the most usage frequency, morning peak, and evening peak – are easily revealed based on other simple statistical methods without a tensor-based approach. Meanwhile, as the non-negative Tucker decomposition approach has been widely used, what is the innovation of the framework?

Third, some modifications could be made to further improve the manuscript:

1. The innovation and the scientific contribution should be clarified earlier in the Abstract. Therefore, readers could quickly capture them.

2. In Section 4.2:” However, as there is no evident pattern for the remaining information dimensions, the numbers of ranks for these modes are then set within predefined ranges: three to eight for day mode and two to nine for both origin and destination modes spatial dimensions”. A wider range is preferred as the optimal rank of core tensor is close to the max value of the predefined range.

3. Fig 4B provides no more information compared with Fig 4A. Merging Figures 4A and 4B is more concise.

4. Fig 8 and Fig 9 should be improved to highlight the patterns.

6. PLOS authors have the option to publish the peer review history of their article (what does this mean?). If published, this will include your full peer review and any attached files.

Reviewer #1: No

Reviewer #2: No

---

## [Author Response · Author response to Decision Letter 0]

28 Jun 2022

Responses to the reviewers' comments have been uploaded as a separate file.

---

## [Decision Letter · Decision Letter 1]

8 Jul 2022

PONE-D-22-13144R1Impacts of the COVID-19 pandemic on the spatio-temporal characteristics of a bicycle-sharing system: A case study of Pun Pun, Bangkok, ThailandPLOS ONE

Dear Dr. Jarumaneeroj,

Thank you for submitting your manuscript to PLOS ONE. After careful consideration, we feel that it has merit but does not fully meet PLOS ONE’s publication criteria as it currently stands. Therefore, we invite you to submit a revised version of the manuscript that addresses the points raised during the review process.

Quality of figures should be improved

We look forward to receiving your revised manuscript.

Kind regards,

Sathishkumar V E

Academic Editor

PLOS ONE

Journal Requirements:

Reviewers' comments:

Reviewer's Responses to Questions

**Comments to the Author**

1. If the authors have adequately addressed your comments raised in a previous round of review and you feel that this manuscript is now acceptable for publication, you may indicate that here to bypass the “Comments to the Author” section, enter your conflict of interest statement in the “Confidential to Editor” section, and submit your "Accept" recommendation.

Reviewer #1: (No Response)

2. Is the manuscript technically sound, and do the data support the conclusions?

Reviewer #1: Partly

3. Has the statistical analysis been performed appropriately and rigorously? 

Reviewer #1: No

4. Have the authors made all data underlying the findings in their manuscript fully available?

Reviewer #1: No

5. Is the manuscript presented in an intelligible fashion and written in standard English?

Reviewer #1: No

6. Review Comments to the Author

Reviewer #1: 1.The results are not very convincing and needs to be improved and I am expecting more data analysis based on the methods proposed.

2.Quality of figures is so important too. Please provide some high-resolution figures. Some figures have a poor resolution.

3.Conclusion briefly summarizes the paper and looks fuzzy. Please rewrite it.

7. PLOS authors have the option to publish the peer review history of their article (what does this mean?). If published, this will include your full peer review and any attached files.

Reviewer #1: No

---

## [Author Response · Author response to Decision Letter 1]

19 Jul 2022

Response to reviewer's comments is uploaded as a separate file.

---

## [Editor Report · Decision Letter 2]

21 Jul 2022

Impacts of the COVID-19 pandemic on the spatio-temporal characteristics of a bicycle-sharing system: A case study of Pun Pun, Bangkok, Thailand

PONE-D-22-13144R2

Dear Dr. Jarumaneeroj,

We’re pleased to inform you that your manuscript has been judged scientifically suitable for publication and will be formally accepted for publication once it meets all outstanding technical requirements.

Kind regards,

Sathishkumar V E

Academic Editor

PLOS ONE
---

## [Editor Report · Acceptance letter]

26 Jul 2022

PONE-D-22-13144R2 

Impacts of the COVID-19 pandemic on the spatio-temporal characteristics of a bicycle-sharing system: A case study of Pun Pun, Bangkok, Thailand 

Dear Dr. Jarumaneeroj:

I'm pleased to inform you that your manuscript has been deemed suitable for publication in PLOS ONE. Congratulations! Your manuscript is now with our production department. 

Kind regards, 

on behalf of

Dr. Sathishkumar V E 

Academic Editor

PLOS ONE